# Body Size and Bite Force of Stray and Feral Cats—Are Bigger or Older Cats Taking the Largest or More Difficult-to-Handle Prey?

**DOI:** 10.3390/ani10040707

**Published:** 2020-04-17

**Authors:** Patricia A. Fleming, Heather M. Crawford, Clare H. Auckland, Michael C. Calver

**Affiliations:** Environmental and Conservation Sciences, Murdoch University, Perth, WA 6150, Australia; crawfh01@gmail.com (H.M.C.); c.auckland@murdoch.edu.au (C.H.A.); M.Calver@murdoch.edu.au (M.C.C.)

**Keywords:** Australia, body condition, diet, *Felis catus*, feral, predation, prey, stray, wildlife, urban

## Abstract

**Simple Summary:**

Predation by cats (*Felis catus*) threatens Australian wildlife. As they rely on their jaws to hold and subdue prey, their body size, skull shape and bite force can reflect an individual’s prey handling ability. Prey less than 100 g are the usual prey of *F. catus* but they have frequently been recorded to take larger prey, and previous studies have suggested that large male cats represent a disproportionate risk to threatened and translocated native wildlife populations. We tested whether a cat’s sex, age, body mass, body condition, and bite force determined the size of the prey they took (prey body mass) especially for those prey that might be ‘dangerous’ or difficult to handle (our subjective assessment of whether animals would be capable of fighting back and would therefore require skill to subdue). Large male cats do indeed represent the greatest risk in that they have greater body mass and bite force that would allow them to handle a greater range of prey. However even small cats were active hunters, and many had taken large or dangerous prey species. The strongest predictor of prey size was the age of the cat, with older cats taking the largest prey.

**Abstract:**

As carnivorans rely heavily on their head and jaws for prey capture and handling, skull morphology and bite force can therefore reflect their ability to take larger or more difficult-to-handle prey. For 568 feral and stray cats (*Felis catus*), we recorded their demographics (sex and age), source location (feral or stray) and morphological measures (body mass, body condition); we estimated potential bite force from skull measurements for *n* = 268 of these cats, and quantified diet composition from stomach contents for *n* = 358. We compared skull measurements to estimate their bite force and determine how it varied with sex, age, body mass, body condition. Body mass had the strongest influence of bite force. In our sample, males were 36.2% heavier and had 20.0% greater estimated bite force (206.2 ± 44.7 Newtons, *n* = 168) than females (171.9 ± 29.3 Newtons, *n* = 120). However, cat age was the strongest predictor of the size of prey that they had taken, with older cats taking larger prey. The predictive power of this relationship was poor though (*r*^2^ < 0.038, *p* < 0.003), because even small cats ate large prey and some of the largest cats ate small prey, such as invertebrates. Cats are opportunistic, generalist carnivores taking a broad range of prey. Their ability to handle larger prey increases as the cats grow, increasing their jaw strength, and improving their hunting skills, but even the smallest cats in our sample had tackled and consumed large and potentially ‘dangerous’ prey that would likely have put up a defence.

## 1. Introduction

In Australia, predation of native fauna by invasive carnivores, including the feral domestic cat (*Felis catus*), is recognised as a ‘Key Threatening Process’ under the Australian Commonwealth *Environment Protection and Biodiversity Act 1999* (EPBC Act). Since its introduction to Australia ≈200 years ago, the domestic cat has become established across the entire continent [1]. Recent estimates indicate that there are 1.4–5.6 million feral cats in natural environments, and another 0.7 million stray (‘semi-feral’) cats in highly modified environments such as urban areas, refuse dumps and intensive farms [1]. It has been argued that feral cats are the single largest threat to biodiversity regardless of ecological context, with 142 native species and subspecies (40 mammals, 40 birds and 21 reptiles) listed as threatened by feral cats [2]. For example, feral cat predation has been implicated in the extinction of at least 22 mammal species in Australia [3]. Marsupials weighing 35 g–5.5 kg (termed the ‘critical weight range’, CWR) from low rainfall areas have been particularly vulnerable to population decline and extinction [4,5,6]; animals within the CWR are ideally-sized prey for introduced eutherian predators, including the feral cat. Predation by feral cats has also been implicated in the failure of several reintroduction programs on mammals, especially those involving species of macropods less than 2.0 kg, dasyurids and peramelids (e.g., Dickman [7]), and feral cat abundance is the best predictor of decline for native conilurine rodents [8]. While there is no doubt that other factors such as habitat clearing, increasing salinity/aridity, altered fire regimes and other introduced pest and predator species all threaten Australia’s native fauna [9], feral cat predation has had the most rapid, dramatic and demonstrable impact on species’ survival [10]. Success in reducing the harmful impact of feral cats requires detailed understanding of their ecology across a broad spectrum of climatic and environmental conditions [11]. An important aspect that needs greater understanding is what factors determine the diet of feral cats? 

There is substantial sexual dimorphism in cats [12]. In Australia, male feral cats generally weigh 3.4–7.3 kg and females 2.5–5.0 kg [13]. It is possible that such differences in body size may influence prey hunted by different sexes, as larger prey are likely to be more ‘challenging’ to hunt or subdue and may require more musculature and skeletal strength for prey restraint. For example, bobcats (*Felis rufus*) show similar body size dimorphism ratio (males average 8.4 kg and females 6.2 kg [14]), and Litvaitis, Clark and Hunt [15] found that male bobcats generally consumed more white-tailed deer (*Odocoileus virginianus*) than female bobcats because males were able to hunt larger prey and defend carcasses from other predators, whereas females preferred smaller snowshoe hares (*Lepus americanus*). While *F. catus* are generally described as having an opportunistic diet [16], there are indications of sex differences in predation behaviour: a review of the profiles of known individual cats responsible for significant impacts on Australian wildlife protection or translocation programs across 20 studies [17] indicated that large male cats ≥3.5 kg were disproportionally responsible for predation events on a range of birds and mammals. Understanding how differences in body size relate to diet would therefore help in understanding influences on diet of feral cats. 

Prey handling ability could also affect the range of prey taken by feral cats. Carnivorans rely on their head and jaws for prey acquisition and ingestion of food, and their skull morphology and bite force can therefore reflect their feeding ecology [18,19]. The hunting behaviour of domestic cats has been thoroughly studied and sequences for specific prey types compared (e.g., rat vs. mouse vs. bird [20]). Typically, the domestic cat ambushes or stalks and pounces on prey, using its forelimbs and claws to pin it to the ground. The spinal cord of prey is then severed when cats place their mouth over the neck and crush their canine teeth between the cervical vertebrae. To deliver the fatal neck bite, cats must orientate their short-muzzled face and eyes close to the prey’s head, risking injury from the prey’s defensive biting and kicking. In response to this defense, cats have developed a ‘play’ hunting sequence in which the cat repeatedly paws and swats the prey [21], regularly picking it up in its jaws and dropping it to the ground again until it tires [22]. Bite force can therefore reflect an important aspect of prey handling ability in cats. Bite force can be estimated from dry skulls using lever models [23,24] allowing analysis of ontogenetic changes in prey handling or direct comparison with diet for the same individuals [25]. Understanding sex differences in bite force, and how bite force develops as cats age and grow, could therefore reflect ontogenetic and sex differences in diet separation. 

In this study, we examined estimated bite forces (as a proxy for prey handling capacity) and the stomach contents for domestic cats (*Felis catus*). To include a broad a range of cat and potential prey our analyses, we included cats collected from rural environments where they were independent of food subsidies from people (which we define as ‘feral’ cats) as well as a smaller sample of those trapped in and around urban areas (which we define as ‘stray’ cats). We examined the relationship between estimated bite force and demographics for individual cats to test the predictions that bite force is greater for males, as well as for larger or older cats. We included sampling location in our analyses, which allowed us to test whether there were differences in estimated bite force for rural feral cats vs. urban stray cats. We then tested whether diet composition or prey size (body mass) was influenced by the cat’s bite force, sex, body mass, age or sampling location.

## 2. Methods

This project was carried out with permission of the Murdoch University Animal Ethics Committee (W2266/09) and adhered to the *Animal Welfare Act 2002*. 

### 2.1. Specimens

A total of 567 culled animals were collected over a period of eight years across almost 100 rural and urban locations across southwest Western Australia (Shires or Suburbs listed in Appendix A). Animals were sourced from licenced animal controllers contracted to various Cities/Shires or natural resource management groups who trapped and euthanised (pentobarbitone overdose or shot) animals that were identified as a threat to endangered native species or that had been reported by the public or local government authority due to nuisance behaviour, welfare concerns, or abandonment [26]. We categorised animals by source location according to the type of anthropogenic development at their immediate point of capture:
Rural: feral cats were trapped and then shot as part of conservation management activities focused around habitats of threatened species (i.e., bushland reserves) or shot while free-roaming on private pastoral properties. These cats live and bre beyond the periphery of dense human settlements and survive independently (i.e., were not fed by people). We did not include animals that had been trapped from around rural refuse sites, as these cats were subsidised to a large extent by human refuse [26]. Urban: stray cats were identified as unowned (‘semi-feral’) cats trapped in urban and peri-urban residential areas, the premises of private and commercial businesses, private small-holdings and fragmented bush reserves within and on the immediate periphery of the metropolitan area (Perth, Western Australia). Enforced from 2013, the *Western Australia Cat Act 2011* requires that, by the age of 6 months, all pet cats must be: (1) desexed, (2) microchipped with sub-dermal ID tag, (3) registered with a local municipal council and (4) wearing a collar with ID and registration tags. Any cat not identifiable as an owned pet trapped within an urban/peri-urban area because of nuisance or welfare concerns, and/or surrendered to a shelter, is therefore considered a stray cat. These stray cats are generally rehomed but may be euthanised if their temperament is not suitable for rehoming. Our stray cats were neither pets nor suitable for rehoming. 

Animal carcasses were stored frozen until they could be analysed in the laboratory. Once thawed, we recorded the sex and body mass (*m_b_*; ±0.01 kg) of individuals. The head length, head-body length and pes length (metacarpal and forepaw of left hindlimb) were measured with a dressmakers’ tape (±0.5 cm). A body condition index (BCI)—reflecting how much heavier-than-average each cat was, accounting for its sex and body size—was calculated for each individual animal as the residual of Log-*m_b_* against body size indicators (head length and head-body length), with sex included as a factor (residual analysis in Statistica 8.0; Statsoft Inc., Tulsa, OK, USA); variability in pes length proved to be too unreliable for inclusion in the BCI calculation.

### 2.2. Diet Analysis

Gastrointestinal tracts were dissected out of carcasses for diet analysis. We analysed stomach contents of *n* = 358 individual cats (excluding animals with empty stomachs). Description of the urban stray cat diet has been published separately [26] but prey composition was reanalysed for comparison to bite force in the present study. Consumed animals were classified as fresh or carrion using tissue friability, smell and/or the presence/absence of maggots; data for animal carrion were not included in the diet composition analysis or analyses of determinants of prey mass because carrion would not pose the same mechanical challenges as live prey capture and handling. Refuse items were classified as sources of food that would have been taken from anthropogenic sources and included food scraps, (e.g., sandwich meats and vegetables, pieces of fruit), paper, plastic, aluminium foil, packing Styrofoam, glass shards, synthetic fibres etc. Trap bait and plant material that was likely to be either deliberately ingested (i.e., green grass) or incidentally ingested during consumption of food off the ground (i.e., small volumes of twigs, leaves, bark) were also identified and excluded from diet analyses for the purpose of this study. For reptiles, we identified prey items down to species where possible, using field guides and expert opinion (see Acknowledgements). We identified mammals and birds to species where possible using either fur [27] or feathers, identifiable body parts and expert opinion. 

We used unpublished trapping data or field guides and other literature (e.g., [28,29]) to determine average body mass of each identified prey species (‘prey size’). Where a cat had consumed a range of species (e.g., one individual had six reptile species present), these data were entered as separate data points, but where a single prey category was evident (e.g., the stomach of one cat contained 30 house mice *Mus musculus*), this was only counted as a single prey size data point. All invertebrates represented one data point (i.e., one prey category) regardless of taxonomic order and are therefore a conservative underestimate of the diversity of invertebrate prey. As well as prey size, we also distinguished potentially ‘dangerous’ prey species—a subjective category based on the authors’ experience of species that could potentially deliver a damaging bite or kick while handled.

### 2.3. Bite Force

Heads were removed from carcasses and soft tissue was removed from skulls by maceration at 30–40 °C. Cats were then aged using tooth eruption patterns, incremental lines of their canine cementum [30], and closure of cranial sutures [Fleming, Crawford, Auckland, Calver, unpublished data]. As shooting was a main form of euthanasia and because the skulls of juvenile cats are less robust than adult skulls (where cranial sutures have fused), a smaller number of skulls (*n* = 268) were available for analysis of bite force compared with diet and demographics. While we made every effort possible to obtain useable data from juvenile cats, they are nevertheless under-represented in our bite force dataset. 

For skull analysis, digital photographs of each skull were taken in lateral, ventral and dorsal views (Figure 1), and measurements (mm) were made using ImageJ software following Damasceno, Hingst-Zaher and Astua [31]. Bite force (in Newtons; N) was calculated using the lever model method according to published formulae [23,24,25,32]. 

Estimated bite force was calculated as the estimated force of the temporalis and masseter/pterygoideus muscle groups (derived from their cross-sectional areas) relative to their centroid distances from the temporomandibular joint (TMJ) and the distance from the TMJ to the canines (the estimated bite force output), which corresponds to the moment or out-lever arm (c). These values were multiplied by two to estimate the estimated bite force (*F*; N) generated by muscle groups on both sides of the skull:
(1)$$\mathrm{Estimated}\;\mathrm{bite}\;\mathrm{force}\;F\;\left(N\right)=2\times \frac{\left(dm\times m\times \sigma \right)+\left(dt\times t\times \sigma \right)}{c}$$
where σ is the isometric muscle stress value of 0.3 N/mm^2^ [32]. 

As the dry skull method has been shown to underestimate bite force, this absolute value was corrected (*F_corr_*) [32] to give the corrected estimated bite force:

(2)$$F_{corr}=10^{\left(0.859\times logF\times 0.559\right)}$$

### 2.4. Statistical Analyses

We compared male and female cat body mass, head-body length and age using Mann–Whitney U Test [33] as the data did not conform to a Gaussian distribution and were not homoscedastic (Levene’s Test). We employed generalised linear modelling (GLM) using the package *lme4* in R [34] to compare estimated bite force (Log-transformed *N*) for each cat (dependent variable with a Gaussian link function) against (1) source location and (2) sex as dummy variables (coded 0 or 1), with the continuous predictor variables (3) body mass (Log *m_b_*; kg), (4) body condition (BCI values) and (5) cat age (Log months). The continuous independent factors were mean-standardised and then we included all combinations of the five independent factors and their first order interaction terms. An Information Theoretic approach using Akaike’s Information Criterion (AIC) was used to identify correlations between bite force and combinations of the five independent variables. We used the *drop1* function in R to compare the global model against variations of the model, excluding each parameter, to support whether each factor was a substantial contribution to model-fit. Akaike model weights (*w_i_*) were calculated for each model in the model-set [35], which were used to weight the standardised beta values (calculated assuming a mean of 0 and standard deviation of 1 for each variable) for each variable across the top models (*Σ**β*·*w_i_*) [36]. 

We used the same modelling approach to compare prey mass (Log *m_b_*; g) against: (1) source location and (2) sex as dummy variables (coded 0 or 1), with (3) estimated bite force (Log N), (4) cat body mass (Log *m_b_*; kg), (5) body condition (BCI values) and (6) cat age (Log months) as continuous predictor variables. 

Differences in diet composition were analysed by Permutational Multivariate Analysis of Variance (PERMANOVA) (*adonis* package) as part of non-metric Multidimensional Scaling (MDS) using the package *vegan* in R [37]. The PERMANOVA [38] included sex and location as the categorical predictor variables, and cat age (months), body mass (kg) and estimated bite force (N) as covariates. This was followed by similarity percentage (SIMPER) analyses. Diet items were grouped as all medium native species (≥500 g), or small native species (<500 g), house mice, black rats *Rattus rattus*, European rabbits *Oryctolagus cuniculus*, birds, reptiles, frogs/fish, invertebrates, grains/animal feed and human refuse. Trap bait, animal carrion and incidental food items (e.g., plant material) were not included in this statistical analysis.

Values are presented as means ± 1 SD unless stated otherwise.

## 3. Results

### 3.1. Demographics

Male cats were 36.2% heavier and 9.2% longer (head-body length) with heads that were 8.9% longer than females, but females in our sample were marginally older than males (Table 1a). Even pregnant females were marginally lighter than the average male body mass. Although rural cats sampled were marginally lighter and shorter than urban cats, only the head lengths were significantly different; there was no significant effect of sample location on the body mass or head-body length between source locations (Table 1b). There was also no significant difference in the ages of cats from these two source locations (Table 1b).

### 3.2. Bite Force

We obtained a complete dataset of bite force as well as age and body measurements for 108 female cats (49.1% feral cats collected from rural locations and 50.9% stray cats from urban locations) and 160 male cats (61.9% feral cats from rural locations and 38.1% stray cats from urban locations). Four models adequately described the variation in bite force in this dataset (likelihood of being the best model of those in our model-set: M1 *w_i_* = 36.4%, M2 *w_i_* = 32.4%, M3 *w_i_* = 16.9%, M4 *w_i_* = 14.3%; Table 2). These models included all predictor variables except for the BCI factor; cat body condition therefore did not contribute to describing the differences in bite force.

The strongest single factor influencing estimated bite force was body mass (model-weighted standardised beta *Σβ·w_i_* = 0.379); heavier cats were capable of generating greater mechanical force than lighter cats (Figure 2a,d). Males had 20.0% greater estimated bite force (206.2 ± 44.7 N, *n* = 168) than females (171.9 ± 29.3 N, *n* = 120) (Figure 2c; *Σβ·w_i_* = 0.202). The interaction between sex and body mass was also included in all the top models: males showed steeper increase in estimated bite force with body mass compared with females (Figure 2b; *Σβ·w_i_* = 0.374). Small males had estimated bite forces less than that of females of equivalent mass, while large males had greater estimated bite forces than females of equivalent mass (Figure 2a).

Age also influenced estimated bite force (Figure 2b,e; *Σβ·w_i_* = 0.094), with older cats having greater estimated bite force than younger animals. Although our sample of stray cats had marginally larger heads, stray cats had smaller estimated bite forces than feral cats. There was also an interaction between source location and animal age (Figure 2e,f); while adults showed little age and bite force difference between rural and urban source locations, young feral cats had greater estimated bite force than young stray cats. A similar relationship was present between location and body mass (Figure 2d), with lighter feral cats having greater estimated bite force than lighter stray cats.

### 3.3. Diet Composition

Only 27 of the 352 cats analysed for diet (7.7%) had consumed carrion (20 sheep *Ovis aries*, three pig *Sus scrofa*, one European rabbit, and three unknown species samples with maggots), all other fauna was freshly consumed or classified as refuse if identified as processed for human consumption (e.g., processed ham).

There was no significant effect of sex (PERMANOVA pseudo-*F*_1,128_ = 0.34, *p* = 0.961), estimated bite force (pseudo-*F*_1,128_ = 1.19, *p* = 0.290), body mass (pseudo-*F*_1,128_ = 0.96, *p* = 0.429) or age (pseudo-*F*_1,128_ = 1.07, *p* = 0.360) on diet composition using 11 broad categories (Table 3). However, diet was significantly different between locations (PERMANOVA pseudo-*F*_1,128_ = 9.23, *p* < 0.001). Similarity percentage (SIMPER) analysis indicated that the main contributors to the differences between rural (feral) and urban (stray) cat diets were the greater amount of refuse present in urban cat diets (SIMPER 35.6% of the difference), and the greater proportion of mice (SIMPER 15.9%), birds (SIMPER 14.7%) and invertebrates (SIMPER 12.0%) in rural cat diets (Table 3). The remaining diet items contributed <10% to the SIMPER analysis (Table 3).

### 3.4. Prey Size (Average Body Mass for Each Prey Category)

We analysed prey size for a total of 611 invertebrate and vertebrate prey items collected from *n* = 294 cats that had fauna present, with an average of 1.72 ± 1.07 (range 1–6) prey categories analysed per individual cat (full list of prey taken by cats given in Table 4). House mice (average *m_b_* 19 g) were the most common prey item recorded, being present in 167 cats, while 107 cats had invertebrates (each estimated as *m_b_* ≈ 0.5 g mass) present in their stomachs. The heaviest prey was black-footed rock wallaby (*Petrogale lateralis*, adult *m_b_* averages 4.05 kg)—body parts including the hindfeet of a juvenile were identified.

For 307 prey items, we had complete body measurements, age and estimated bite force for the cat that had consumed them. There was no single best model to describe prey body mass data. Thirteen models had a ΔAIC < 2; none of these 13 models had particularly strong predictive power although all models returned a significant coefficient of determination (*r*^2^ < 0.038, *p <* 0.003). Cat age was the strongest factor, included in 10 of 13 of these top models (*Σβ·w_i_* = 0.101, Figure 3c,d). Bite force was represented in seven of 13 models (*Σβ·w_i_* = 0.066, Figure 3e,f) and cat mass was present in six of 13 (*Σβ·w_i_* = 0.044, Figure 3g,h). Sex (4/13 models, *Σβ·w_i_* = 0.029), source location (3/13 models, *Σβ·w_i_* = 0.012) and body condition (1/13 models, *Σβ·w_i_* = −0.002) each had low explanatory power to describe prey body mass. Prey species taken by feral (average prey *m_b_* 279 ± 655 g, *n* = 545) and stray (190 ± 543 g, *n* = 94) cats were not statistically significantly different in terms of their body mass (Mann–Whitney U test *Z* = 1.34, *p* = 0.179). There was also no difference in the average prey mass taken by male (284 ± 641 g, *n* = 360) and female (247 ± 646 g, *n* = 280) cats (Mann–Whitney U test *Z* = 1.52, *p* = 0.129). 

Black rats, a number of parrots, venomous snakes and varanids (‘monitor’), as well as the European rabbit, brushtail possum and the black-footed rock wallaby, were subjectively identified as potentially ‘dangerous’ prey that could be considered more difficult for cats to handle (Table 4). There were no significant relationships for the presence of ‘dangerous’ prey items by cat age (see the black lines on Figure 3c, *r*^2^ = 0.033, *p* = 0.064), by estimated bite force (Figure 3e, *r*^2^ = 0.003, *p* = 0.639), or by cat body mass (Figure 3g, *r*^2^ = 0.013, *p* = 0.171).

## 4. Discussion

Review of previous studies has suggested that large male cats represent a disproportionate risk to threatened and translocated native wildlife populations in Australia [17]. To test this observation tested whether a cat’s sex, age, body mass, body condition, and bite force reflected the size of the prey they had consumed. Males were 36.2% heavier and had 20.0% greater bite force than females. Males therefore attain greater body mass and bite forces that would allow them to handle a greater range of prey. However, the strongest predictor of prey size was the age of the cat, with older cats taking the largest prey on average. Older cats had greater estimated bite force and would also have an experience advantage over younger animals, as an individual’s hunting skills would improve with repeated exposure to a broadening range of prey types [39]. None of the factors that we tested were predictors of the body mass of ‘dangerous’ prey items that had been consumed.

### 4.1. Bite Force Anatomy

Felids vary markedly in body size, with skull size and structure strongly correlated with bite force and maximum prey size taken [40,41]. Bite force is therefore a useful proxy for prey handling ability in these carnivorans, revealing differences in potential diet. Cat jaw mechanics follow the general carnivore pattern of having a high coronoid process, hinge-like jaw condyle to improve mechanical leverage of the temporalis muscle, and a tooth row at the same level of this condyle [40]. The masseter and pterygoideus muscles play a crucial role in adduction and lateral stabilisation of the jaw during subduing and handling of prey. Among felids, species of the domestic cat lineage show a broader face, and the ratio of their zygomatic arches breadth to skull length is significantly larger than in most other Felidae species [40]. Furthermore, *F. catus* possess a relatively broad skull with wide braincase and robust cheek bones—typically associated with the allocation of strong masseters [40]. Felids have a short rostrum and, consequently, a short jaw out-force moment arm—such adaptation typifies a skull with large jaw muscles that facilitate mechanical closure and substantial grip, such as is required for subduing moving prey. This robust anatomy contrasts with skulls that have a longer snout that would enable higher velocities at the canine, a characteristic of predators specialising in relatively small, more agile prey and those that use a pounce-pursuit or ambush hunting style [25,42].

Body mass was the strongest single factor influencing estimated bite force in our sample of *F. catus*. The body mass values that we recorded (overall average of 3.08 ± 1.24 kg for our sample of *n* = 568) are within national ranges (e.g., 1.50–7.30 kg [43,44,45]). Our maximum body masses (males 6.7 kg, females 4.9 kg) were smaller than extremes documented by other studies (e.g., [43,44,45]), and anecdotal records for large feral cats in Australia [46], which, if they can be verified, are alarming. For example, in Gippsland, Victoria, a hunter claimed to have shot a giant cat with head-body measurement of 1.6 m plus a 0.6 m tail [47]. If feral cats are increasing in size over generations, then the predatory impact of such giant cats on native fauna may also increase. Taking this argument a step further, Dickman, Legge and Woinarski [48] have identified the risk of importing felid genetics into Australia that could contribute to larger feral cats. Overseas, *Felis catus* has been deliberately cross-bred with other felid species under captive conditions to produce hybrid ‘designer breed’ cats with characteristics that are deemed to be desirable by the domestic pet trade [48]. For example, ‘savannah cats’ are a cross between *Felis catus* and the African serval *Leptailurus serval*. Servals are extremely agile [49] and efficient hunters (49% of hunting attempts yield prey), and while their diet generally comprises prey items <200 g, they can also take medium-sized mammals, birds and reptiles [50]. Savannah cats have inherited the serval’s long legs and ability to jump several metres in a single bound [49]. Weighing 2–4 times the average mass of *F. catus* (males can weigh 8.0–11.0 kg; females are slightly smaller [48]), savannah cats could therefore pose a threat to a greater range of Australian native species. In recognition of this threat, the Australian government banned the importation of the savannah cat in 2008 [48]. 

Sexual dimorphism in skulls is common in felids [40]. The cat data we collected conformed with this pattern, with male cats being 36.2% heavier than females (males 3.50 ± 1.34 kg, females 2.57 ± 0.89 kg) and having 20.0% greater bite force. While small males had estimated bite forces somewhat less than that of females of equivalent mass, males developed disproportionately greater estimated bite force as they grew, such that heavier males had greater estimated bite forces than females of equivalent mass. Males should therefore have a greater range of prey sizes available to them, although we note that there were no sex differences in overall diet composition.

Estimated bite force increased with age in *F. catus*. An increase in bite force measures with age has similarly been reported in a range of other species (e.g., [25,51,52]). Older *F. catus* should therefore have greater prey handling ability, independent of their skills and experience. Anecdotally, we had one large male cat (#362, 4.24 kg, BCI 18.0% heavier than predicted from his body size) that had no teeth at all—they had all broken off. Closure of cranial sutures indicated that he would have been at least 6 years of age [Fleming et al. unpublished data], but was likely older than this (without teeth we could not age the animal using incremental cementum lines). The cat had 10 house mice and domestic chicken in its stomach and so, despite lacking teeth, it was clearly still a successful predator.

We also recorded differences in bite force according to their source location, with feral (rural) cats having greater estimated bite forces compared with stray (urban) cats, while there was no difference in body mass by source location. The difference was most marked for younger and smaller cats, where feral cats had greater estimated bite force than their stray counterparts. Handling live prey would influence the development of bite force, and differences in diet composition (see next section: greater proportion of live prey present in the diets of feral cats) are therefore likely to be driving these differences in bite force.

Cat body condition did not contribute to the differences in estimated bite force, suggesting that even nutritionally-stressed cats with lower body condition are capable of delivering a similar estimated bite force as relatively heavier individuals. Bite force is estimated from the skulls, looking at the area of muscle attachment. This is ‘retrospective’ in the sense that it represents past condition—the cat had to have had the muscles at some time to need that area of support. Therefore, current condition is not the key predictor of bite force—past condition, rather, or perhaps condition in adolescence or young adulthood, is likely to be more important. 

### 4.2. Diet Composition

Cats in our sample had consumed a broad range of prey, including invertebrates, birds, reptiles and small and medium-sized mammals. Despite the differences in bite force and body mass that we recorded, the anatomy of cats in our sample did not appear to influence their diet composition. The only factor that affected diet composition was the source location of these cats. 

The greatest difference in diet of feral (rural) and stray (urban) cats was due to the abundance of live prey for feral cats (house mice, rabbits, reptiles, and invertebrates) compared with human refuse for stray cats. These findings likely reflect differences in the foods available in each environment. For example, an average of 18.8% ± 30.0% of stomach contents volume for our stray cat sample was made up of refuse—mostly human food scraps and plastic, paper and foil (more detail has been published elsewhere [26]). Other studies similarly report various forms of ‘food scraps’ in cat diets (e.g., [53,54,55,56]). These data suggest that cats living close to human habitation modify their diet, possibly in relation to either prey availability or preference for easily obtained scavenged refuse (e.g., old sandwiches, meat trays) that requires less jaw strength to access and ingest. 

While both stray and rural cats consumed human refuse, very few ingested animal carrion (7.7%, total *n* = 27 ate sheep, pig, rabbit or unknown carrion, 9 peri-urban cats, 18 rural; carrion data were not included in our diet analyses). Other authors have also reported that carrion is rarely taken by feral cats (e.g., [57]). Carrion was readily taken by red foxes sourced from the same agricultural environment as our rural cats [25], so was clearly available. The cat’s propensity to hunt live prey even when carrion is available may therefore threaten native fauna still surviving in agricultural areas. 

There was no significant difference (*p* = 0.179) in prey body mass taken by feral (279 ± 655 g, *n* = 545) or stray (190 ± 543 g, *n* = 94) cats in this study. Diet data in the present study may be limited by prey availability at our sample locations; we note that we had also removed carrion from our analyses. By contrast with our findings, Bateman and Fleming [16] analysed data presented by Pearre and Maass [58] and found that cats sampled from sites close to human habitation (farms, suburban and urban studies) take significantly smaller prey (23.2 ± 8.3 g; *n* = 16 studies) than cats in rural areas (72.6 ± 92.1 g, *n* = 28 studies). The broader prey base available across different studies at different locations (and perhaps also the inclusion of carrion) may be a better test of this location difference in prey size than for our present study.

In terms of Australian mammal prey, introduced rabbits and house mice are predominant in diet of cats from semi-arid and some arid habitats, while marsupials are predominant in diet of cats from temperate forest, urban and suburban habitats (reviewed by [7,11]). Cats are noted rodent specialists (e.g., Dickman and Newsome [21]) and rodents were the most common prey taken by cats in our sample. Just over half the cats in our study (56.8% of those that had consumed fauna) had consumed house mice prior to trapping, with rural cats taking the largest numbers (maximum 30 mice in one cat’s stomach). By contrast, only 7.1% of cats had consumed black rats. In comparison to mice, rats are large and aggressive and take skill to dispatch [59] (matching our definition of ‘dangerous’ prey for the present study). For example, two studies on the interactions between rats and cats cohabiting in alleyways in Baltimore, USA, found that stray cats preyed on juvenile but not adult rats and would scavenge refuse despite presence of rats [60,61].

Where these introduced animals have become well established [62], European rabbits are common prey in the diet of cats across many parts of Australia (e.g., [21,53,63,64], reviewed by Doherty et al. [11]). There is some suggestion that larger male cats are able to take larger rabbits (e.g., Hart [65], but see Catling [64] who found no difference in the representation of rabbit in the diet of ‘immature’ females <2.2 kg, males <3.5 kg and ‘mature’ cats). Anecdotal data suggest that cats in some ‘rabbit-infested’ inland regions may be larger than those in rabbit-free areas in northern and eastern Australia (P. Wagner, pers. comm. cited by Dickman [7]), perhaps indicating selection for larger body size, or greater longevity due to access to prey [7]. Although not common across our southwest Western Australian study sites, rabbits were evident in low abundance in the stomachs of cats we sampled: averaging 3% ± 15% by stomach contents volume, or 57% ± 33% of stomach contents volume for the 34 cats that had eaten rabbit. Prey availability clearly plays an important role in diet for feral and semi-feral cats, as has also been noted for pet cats across Australia (see short overview in Grayson and Calver [66]), which may account for variation in prevalence of rabbits in the diet of cats [11].

Feral cats include a substantial number of birds in their diet, with an estimated toll of 272 million Australian birds per year (95% confidence interval: 169–508 million) [67]. Birds were common in the diets of the cats we studied with several cats consuming >1 bird species or individual (one cat ate four bird species, three cats ate three species, and one cat ate 11 individuals of two species). In reviewing the traits of birds predated by cats, Woinarski et al. [68] reported increased cat predation risk for birds that nest or forage on the ground. Our data support this finding, with evidence of cats having consumed at least seven parrot species, many of which are ground-foraging species. Notably, 11.2% of cats had feathers or body parts of Australian ringneck parrots present in their stomachs. These parrots are gregarious and flighty so hunting them on the ground takes stealth. Other ground-foraging species were also common, including pigeon, button- and stubble-quail species, with one cat having consumed 10 baby stubble-quails (in addition to a small passerine and two reptiles). 

Cat predation has been recorded for about one-quarter of described Australian reptile species [69]. A broad range of reptile species were similarly recorded as prey in both rural and urban cats in the present study. Cats had consumed at least 23 reptile species, including monitors, dragons, skinks, geckos, snakes and ground burrowing blindsnakes. Some of these species would be dangerous to handle. For example, monitor lizards (e.g., *Varanus gouldii*, measuring up to 1.4 m long and weighing up to 6.0 kg) grow to be large and aggressive; they are fast runners with strong legs and sharp claws, and use their whip-like tail as a defensive weapon. In our study, three of the seven cats that had eaten monitor lizards were aged just 5–10 months and weighed 1.4–4.0 kg. Six cats in our study (weight range 2.3–4.8 kg) had eaten venomous snakes, with the youngest cat only 10 months of age. McGregor et al. [70] video recorded rural cats skilfully catching a venomous adult western brown snake (*Pseudonaja mengdeni*, ≈1.2–2 m in length) and immediately crushing its head to subdue the ‘dangerous end’. Cats are therefore clearly a threat to even venomous snakes. Our data also suggests that even young cats could threaten venomous and non-venomous reptiles.

We recorded the consumption of frogs by only two cats. This finding conforms with studies that have found frogs to be an uncommon prey item, except for island cat populations [71]. Both cats were from rural locations. One had a single western banjo frog in its stomach while the other had 30 juvenile endemic Kunapalari frogs of identical size that were probably hunted at a single breeding location (e.g., farm dam). Such large numbers of prey suggest an opportunistic hunting strategy that could impact local populations of frogs.

Invertebrates are normally viewed as a minor supplementary food source for cats (e.g., [53,57]) except for on islands or in arid/semi-arid areas where invertebrates are readily available and where other prey taxa may be limited or seasonal [71,72]. However, we recorded substantial amounts of invertebrates in the diets of both stray and feral cats. Up to 51 g of invertebrates (actual measurements of the mass of invertebrates recorded from the stomach) were eaten by 31% of cats (body mass range 0.5–5.8 kg), of both sexes and across locations, with grasshoppers, beetles and centipedes most commonly ingested. Grasshoppers and centipedes were similarly the most frequent invertebrates recorded from feral cat stomachs from Eastern Musgrave Ranges in arid South Australia [44], although they rarely made up the majority of stomach contents. Invertebrates can be highly nutritious, especially gravid females that have large body mass and lipid concentrations in the developing eggs [73]. Additionally, although cats can survive without access to free-standing water [74], it is likely that invertebrates are an important source of water, fat and protein for cats when other prey is scarce (similar to red foxes [75,76,77]). This may explain why, regardless of bite force, 36.4% of cats had consumed invertebrates. The importance of invertebrates as prey likely varies with location and season, in keeping with the understanding of cats as opportunistic predators.

### 4.3. Prey Size

Prey body mass was positively correlated with cat age as well as their bite force and body mass. These relationships were not strong, however, with models each explaining less than 4% of the variability in prey body mass. This is due to the wide range of prey sizes taken by all cats. Even small cats had large prey items in their stomach contents; for example, two of the cats that predated European rabbit (≈1.8 kg) weighed 1.4 and 1.5 kg respectively. Furthermore, large cats had taken many small prey, with 20% of cats weighing more than 5.0 kg having consumed invertebrates, our smallest prey size category (≈0.5 g).

Felidae species ≥10 kg can take prey larger than themselves, while species <10 kg tend to take prey that are smaller than themselves [40]. The average mass of prey taken by cats in our study was 379 ± 740 g for 434 prey items (excluding invertebrates). This is around 12.4 % of the average body mass for our sample of cats and therefore similar to predicted prey range based on their body mass (e.g., 13% [78], 11% [79]). Paltridge et al. [57] similarly report common prey in the range of 10–350 g. Animals within the critical weight range (35 g–5.5 kg) are ideal weight range for cat prey [5,80]. Such small prey species would be unlikely to pose a challenge to the cat in terms of the bite force required to handle the prey.

Larger prey have also been recorded in the diets of feral cats in Australia [80], albeit less frequently [7,11,80]. For example, Dickman [7] records bird prey up to 3.5 kg on islands as exceptions. Compared with smaller prey, medium-sized marsupials are likely to pose a greater hunting challenge because their body size represents a reasonable match to an average cat. We recorded the remains of a juvenile black-footed rock wallaby in the stomach of one cat trapped from around a protected rock wallaby population. The female cat that had eaten the wallaby weighed 3.5 kg, a near match for a juvenile wallaby. In addition to their body size, marsupials such as rock wallabies (*Petrogale* spp.) also escape predators using propulsive hopping locomotion and are capable of delivering injurious kicks. Despite their size and challenges with handling them, however, there is mounting evidence that even medium-sized marsupials are taken by feral cats as prey (Table 5). For sexually dimorphic prey species (where females are smaller), females and juveniles are more vulnerable to cat predation due to their smaller body size [81], while juveniles are also predator-naïve and females with pouch-young are more encumbered [82]. Even if it is relatively infrequent, predation by feral cats therefore represents a significant conservation issue for populations of medium-sized threatened species.

Cats appear to learn how to take large prey, with individuals becoming specialised to particular prey [21], and therefore the incidence of specific larger prey species being taken can increase in a cat’s diet over time. Hardman, Moro and Calver [87] present evidence showing that once one translocated rufous hare-wallaby (*Lagorchestes hirsutus*) was killed, predation continued until each population was eliminated; in many cases this extirpation was within a matter of days, suggesting that individual cats quickly became specialised as hunters of these mammals. Similarly, Gibson et al. [82] report that trapping of specific individual cats (no mention of their body mass) resulted in the cessation of predation on rufous hare-wallabies. Our data supports these field observations, with a positive correlation between prey body mass and cat age that could support learned skills in prey capture as cats age. Feral cat trapping programmes are therefore essential to preserving even medium-sized marsupial populations. 

### 4.4. Other Considerations—Bite Force and Fighting/Territorial Mating Behaviour 

Increases in ‘bite performance’ [89], which presumably includes bite force, improves territorial defence and therefore mating possibilities in lizards [89,90,91], rodents [92] and lemurs [51]. Cats are territorial and will fight other cats using claws and biting [93,94], and cats with greater bite forces could therefore be advantaged in physical confrontations. Male cats also restrain females during copulation by biting the back of the neck [95], taking advantage of immobility induced by the ‘scruff reflex’ [96] and protects the male from retaliatory aggression from the female. Bite force in cats may not be solely a product of prey selection, therefore, with sexual selection also possibly playing a role. 

### 4.5. Limitations of This Study

There are a few caveats to interpreting diet analyses based on stomach contents of these cats. 

First, the diet for our animals represents a snapshot in time—our diet data are only a measure of what each individual cat had eaten in the previous ≈12–24 h and would also be influenced by prey availability and if cats were trapped in cages for a period. There is reasonable evidence to suggest that individual cats become specialised on particular prey [17,21,97,98], however, which may lend weight to the interpretation of diet from a single sample as reasonably reflective of what that cat might eat on an ‘average day’. 

Second, some prey items identified from hair or body parts could be juveniles, and therefore the species’ average adult body mass values that we compared our data against would need to be considered with caution. The black-footed rock wallaby consumed by a cat was identified as juvenile (although adult rock wallabies are also predated by feral cats [44]). Identifying the size of prey from stomach contents can be informative. For example, Catling [64] identified that small rabbits (estimated to be <50 days old) comprised 73% of the diet of both mature and immature cats, but adult rabbits comprised only 8% of their diet.

Third, identifying whether prey are taken live should always be considered. Although we retrieved whole or parts of animals, we also identified prey species from hair or feathers. Arguably, some of these could represent prey that had escaped capture (e.g., birds). Many authors report that carrion is rarely consumed by feral cats (e.g., [57,80]), although this may depend on alternative food availability. This avoidance of carrion is behind the difficulty reported in baiting feral cats and the extensive work undertaken in developing poison baits that are attractive, especially at times of high food availability [99,100,101]. We similarly found that carrion (defined as consumption of part of a decomposing carcass, with or without fly maggots, as opposed to scraps of discarded human foods classified as refuse) was uncommon—only 27 cats had consumed sheep, pig or rabbit carrion (<8% of the 352 cats analysed for diet). As we are confident that we separated carrion from kills, we know that the prey were hunted and killed as live animals. 

Finally, female cats bring back prey for their young which means that small cats would have access to larger prey than they could capture themselves. From 4 weeks post-partum, mothers bring their kittens a range of live and dead prey so that they can ‘play’ with different foods and develop hunting sequences (e.g., pouncing, grappling; species-specific hunting sequences needed to kill rats vs. bird etc. [102,103]). Kittens are generally weaned from maternal care and start hunting independently by the age of 3–4 months [39,97], but weaning can occur earlier in resource-poor environments where the lactation period may be reduced [103]. Our sample therefore included very few cats that would have still been dependent on their mothers for food. The prey detected in the four kittens aged <4 months in our study included a single mouse, and invertebrates in three kittens, and likely reflects this maternal provision (or opportunistic consumption). However, older cats were likely to be capturing their own live prey. Sixteen cats were under 6 months of age (estimated to be either 4 or 5 months and possibly still weaning). While most had ingested invertebrates, four of these had consumed vertebrates: a blindsnake, a black-headed monitor, a brushtail possum, and a domestic chicken—two of which we subjectively classified as ‘dangerous’ prey. Our sample also included 155 juvenile cats aged between 6 and 10 months of age which should all have been hunting independently. Trap bait or human refuse was present in the stomach of 81 of these cats (52.2%) and probably reflects opportunistic consumption of available ‘free’ foods. The remaining 74 cats consumed a wide range of prey (37.8% of cats 5–10 months ate 122 vertebrates; 21.6% ate invertebrates), with up to seven and eight vertebrates in two stomachs each respectively. Notably, many of the prey species taken by these young cats were large and/or dangerous. For example, one cat ate a highly venomous western brown snake, three cats ate monitor lizards, six ate parrots, five consumed black rats, and three consumed brushtail possums. Such dangerous prey require considerable skill to capture and subdue without injury to the cat, so their consumption illustrates just how quickly cats become adept hunters following weaning. 

## 5. Conclusions

The availability of prey is presumably the strongest determinant of stray and feral cat diets, which will opportunistically feed on common or most easily captured prey (reviewed by [11]). Various studies similarly indicate that learned behaviour influences prey taken by cats (e.g., [21,87,97]). Our data adds to this list of factors influencing cat diet by revealing that the range of prey accessible to cats is also influenced by their anatomy. Large male cats do indeed represent the greatest predation risk for native wildlife species in that their body mass and bite force would allow them to access a greater range of prey species. The threat posed by smaller cats is not to be ignored, however, as even small cats in our study had been active hunters, and many had taken some surprisingly large prey species. 

The strongest predictor of prey size in our study was the age of the cat, with older cats taking larger prey on average. We note that age is also the strongest factor correlated with relative body condition for a sample of *n* = 79 male stray cats (there was no obvious relationship for *n* = 77 females [26]). Other evidence of increased hunting success with age comes from cheetah (*Acinonyx jubatus* [104]) and spotted hyena (*Crocuta crocuta* [105]). Success in hunting with age has also been documented in raptors (e.g., northern goshawk *Accipiter gentilis*), where older birds have improved hunting success, increased provisioning, and therefore greater breeding success [106]. While male cats do not provision their young or their mates, greater hunting success would nevertheless influence their own fitness, while greater hunting success in older females would benefit themselves as well as their young. Two of the oldest feral cats in our sample we estimated to be ≈14 and ≈16 years of age (both were females; notably one was lactating and the other pregnant). One had consumed livestock feed (demonstrating opportunism in diet) while the other had both bird and reptile remains in its stomach. Another old male (unknown age) was demonstrably a successful mouse hunter, despite lacking all his teeth. Age gives a combination of size and experience that both facilitate successful hunting of the greatest range of prey sizes and taxa. Consequently, age and body mass together describe a cat that poses the most substantial threat to native species. 

## Figures and Tables

**Figure 1 animals-10-00707-f001:**
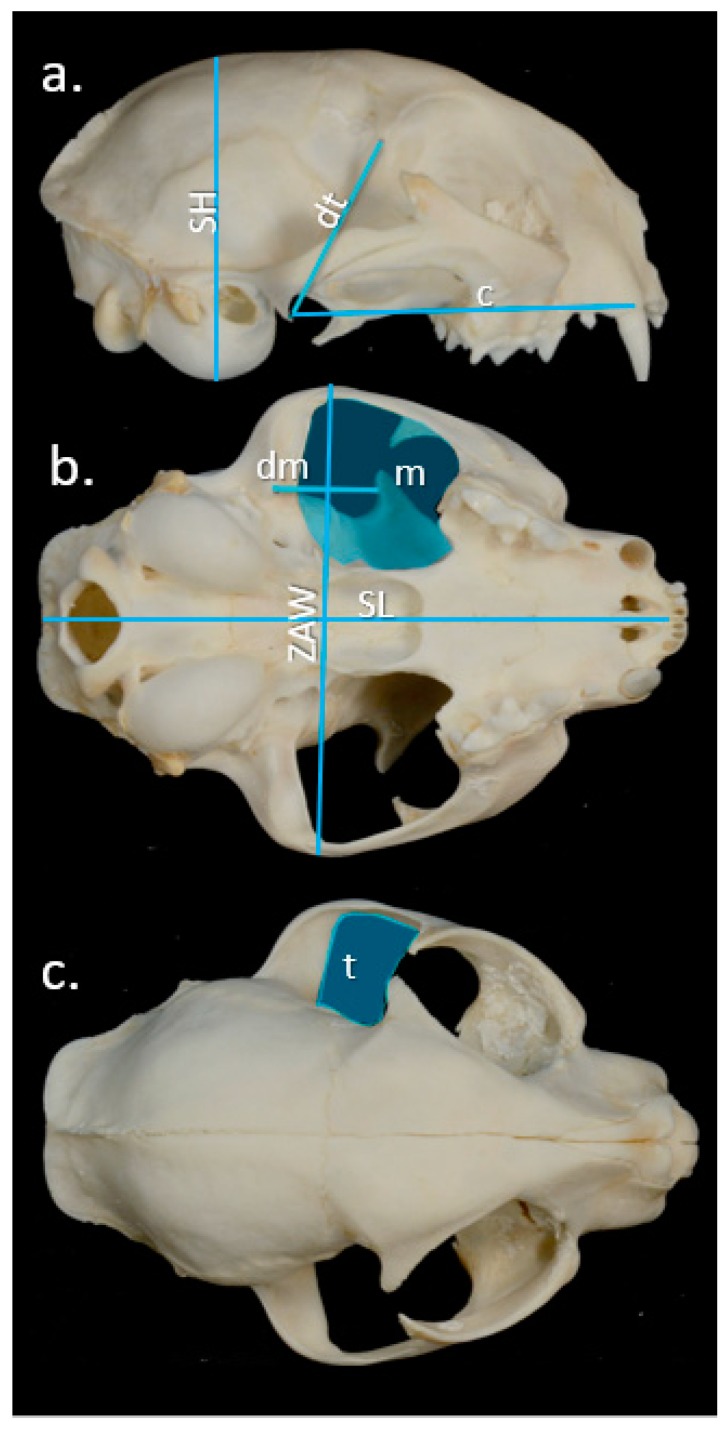
(**a**) Right lateral, (**b**) ventral and (**c**) dorsal view of a cat (*Felis catus*) skull. Distances recorded for estimation of bite force: (**a**) **SH** skull height, **dt** the distance between the centroid of the temporalis and Temporo-Mandibular Joint (TMJ) and **c** distance between the canine and TMJ; (**b**) **ZAW** zygomatic arch width, **SL** ventral skull length, **dm** distance between the centroid of the masseter muscle and the TMJ, and the cross-sectional areas of the masseter muscle/pterygoideus muscle (**m**); and, (**c**) the cross-sectional area of the temporalis muscle (**t**).

**Figure 2 animals-10-00707-f002:**
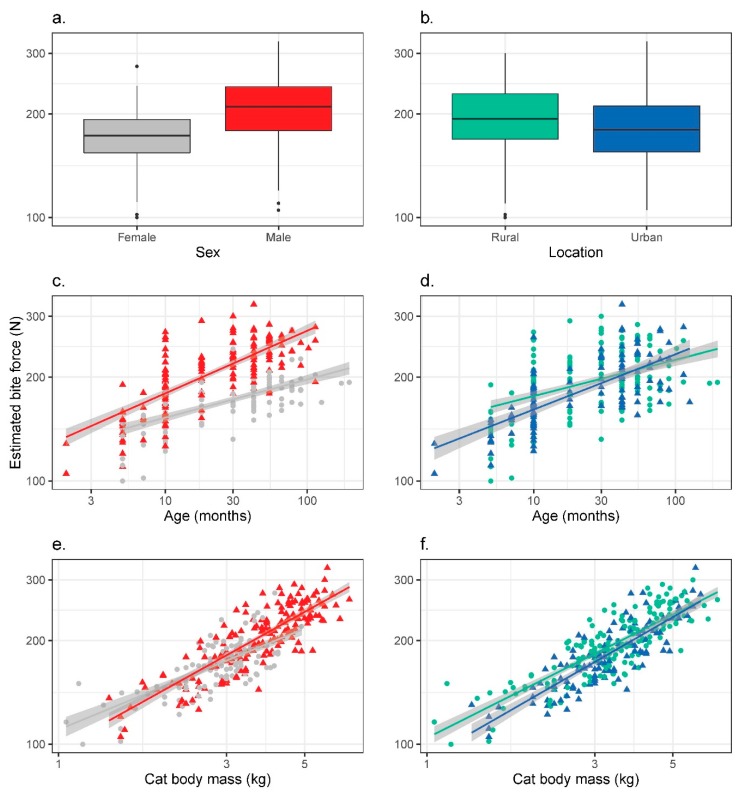
Estimated bite force (note Log-transformed axis) of cats (*Felis catus*) from southwest Western Australia shown by (**a**) sex (left-hand panel; males = red triangles *n* = 160, females = grey circles *n* = 108) or (**b**) source location (right-hand panel; rural = green circles and urban = blue triangles). Estimated bite force shown against age (**c**,**d**) and cat body mass (**e**,**f**).

**Figure 3 animals-10-00707-f003:**
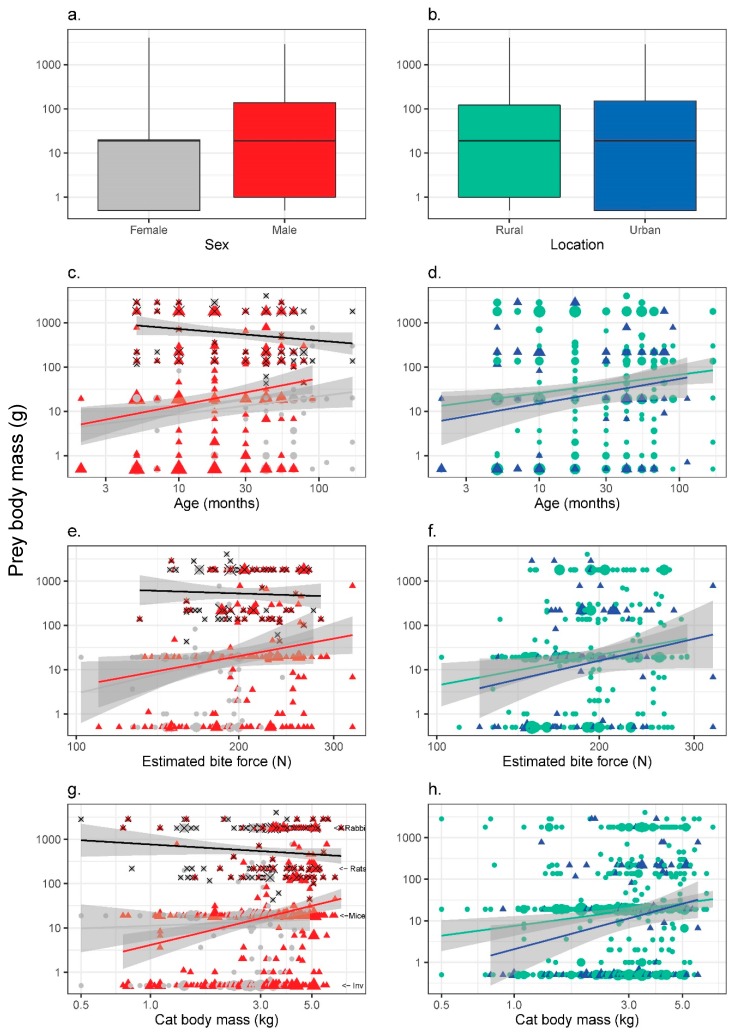
Prey body mass (average body mass for an adult of that species) consumed by cats (*Felis catus*) from southwest Western Australia shown by (**a**) sex (left-hand panel; males = red triangles and females = grey circles) or (**b**) source location (right-hand panel; rural = green circles and urban = blue triangles). Prey body mass shown against age (**c**,**d**), estimated bite force (**e**,**f**) and cat body mass (**g**,**h**). Note Log-transformed axes. Due to overlapping data, symbol sizes represent relative numbers of prey items. Diet was determined for some small cats that we could not determine bite force for (and therefore the x axis has a lower minimum value compared with Figure 2). The most common prey items are indicated in (**g**): European rabbits (*Oryctolagus cuniculus* 1.8 kg), black rats (*Rattus rattus* 220 g), house mice (*Mus musculus* 19 g), and invertebrates (all grouped as 0.5 g). Prey that might be perceived as requiring greater hunting skills (‘dangerous’ prey) are indicated with a black cross and the black regression lines in left-hand panels.

**Table 1 animals-10-00707-t001:** Demographics and significance testing for stray and feral cats (*Felis catus*) presented by (a) sex and (b) source location in southwest Western Australia.

**(a) Sex**	**Males**	**Females**	**Mann–Whitney U test**
Body mass	3.50 ± 1.34 kg, *n* = 308, max 6.7 kg	2.57 ± 0.89 kg, *n* = 260, max 4.9 kg	*Zadj* = 8.72, *p* < 0.001
Pregnant		3.36 ± 0.51 kg, *n* = 24	
Non-pregnant		2.49 ± 0.88 kg, *n* = 236	
Head-body length	551 ± 84 mm, *n* = 299	504 ± 74 mm, *n* = 247	*Zadj* = 7.36, *p* < 0.001
Head length	112 ± 14 mm, *n* = 288	103 ± 13 mm, *n* = 235	*Zadj* = 7.01, *p* < 0.001
Age	2.42 ± 2.05 years, *n* = 154	3.21 ± 2.88 years, *n* = 105	*Zadj* = 1.98, *p* = 0.047
**(b) Location**	**Rural (feral cats)**	**Urban (stray cats)**	**Mann–Whitney U test**
Body mass	3.05 ± 1.26 kg, *n* = 419, max 6.7 kg	3.14 ± 1.20 kg, *n* = 148, max 6.0 kg	*Zadj* = 0.56, *p* = 0.575
Head-body length	527 ± 83 mm, *n* = 398	538 ± 82 mm, *n* = 148	*Zadj* = 1.13, *p* = 0.257
Head length	106 ± 15 mm, *n* = 375	110 ± 12 mm, *n* = 148	*Zadj* = 2.60, *p* = 0.009
Age	2.87 ± 2.56 years, *n* = 143	2.58 ± 2.30 years, *n* = 116	*Zadj* = 1.11, *p* = 0.266

**Table 2 animals-10-00707-t002:** Summary of generalised linear modelling testing for factors influencing bite force in stray and feral cats (*Felis catus*) in southwest Western Australia.

Model		df	AIC	Delta AIC	*w_i_*
1	BF ~ Cat *m_b_* * Sex + Location * Age	8	−828.69	0.00	0.364
2	BF ~ Cat *m_b_* * Sex + Location * Age + Location * Sex	9	−828.45	0.23	0.324
3	BF ~ Cat *m_b_* * Sex + Mass * Location + Location * Age	9	−827.15	1.54	0.169
4	BF ~ Cat *m_b_* * Sex + Location * Age + Sex * Age	9	−826.81	1.88	0.143

BF bite force was log-transformed (N); Cat body mass (*m_b_*) was log-transformed (kg); Age was log-transformed (months).

**Table 3 animals-10-00707-t003:** Similarity percentage (SIMPER) analysis comparing the diets of *n* = 256 rural (feral) and *n* = 102 urban (stray) cats (*Felis catus*) collected across southwest Western Australia.

Diet Category	Contribution to Difference in Diet (SIMPER %)	Rural = Feral Cat Mean ± 1 SD Percentage Volume	Urban = Stray Cat Mean ± 1 SD Percentage Volume
Refuse	35.6	9.25 ± 25.72	18.83 ± 30.02
House mouse *Mus musculus*	15.9	28.56 ± 39.57	1.49 ± 8.10
Birds	14.7	10.03 ± 25.15	5.32 ± 19.38
Invertebrates	12.0	4.32 ± 14.25	1.24 ± 4.54
European rabbit *Oryctolagus cuniculus*	5.7	4.91 ± 18.77	0.49 ± 4.95
Reptiles	5.2	2.85 ± 11.28	0.01 ± 0.10
Black rat *Rattus rattus*	4.7	2.36 ± 13.82	3.97 ± 17.20
Frogs/Fish	2.4	1.03 ± 9.53	0.15 ± 1.49
Total medium native mammal ≥500 g	2.1	0.97 ± 9.08	0.90 ± 8.91
Grains/Animal feed	1.8	2.76 ± 13.38	0.00 ± 0.00
Total small native mammal <500 g	<0.01	0.07 ± 1.09	0.00 ± 0.00

Medium-sized native mammals included black-footed rock wallaby *Petrogale lateralis*, common brushtail possum *Trichosurus vulpecula hypoleucus* and quenda *Isoodon fusciventer*. Small native mammals included western bush rat *Rattus fuscipes fuscipes*, fat-tailed dunnart *Sminthopsis crassicaudata* and the western pygmy possum *Cercartetus concinnus*.

**Table 4 animals-10-00707-t004:** List of prey species identified from stomach content analysis of *n* = 294 feral and stray cats (*Felis catus*) from southwest Western Australia that had fauna present in their stomachs. The number of cats eating each prey item does not equal the total number of species eaten as many cats ate >1 species. Except for invertebrates (which were all assigned an arbitrary mass value of 0.5 g, regardless of whether or not we could identify them), analysis of prey mass excluded prey items that could not be identified sufficiently to ascribe a mass to the item. The average body mass for each prey species (carried out for *n* = 611 prey items) represents an approximation of adult size from various sources in the literature. Prey were distinguished as potentially ‘dangerous’ prey species—a subjective category based on the authors’ experience of those species that could potentially deliver a reasonable bite or kick while being handled.

Prey Species	Species Name	‘Dangerous’	Avg. *m_b_* (g)	Eaten by No. Cats
**Invertebrates**				
(all orders/class/clade grouped)		N	0.5	107
Cockroaches	*Blattodea*			
Beetles, weevils	*Coleoptera*			
Wasps, fly larvae	*Hymenoptera*			
Moths, moth larvae	*Lepidoptera*			
Dragonflies	*Odonata*			
Grasshoppers, crickets, mole-crickets	*Orthoptera*			
Scorpions	*Scorpiones*			
Spiders	*Araneae*			
Earthworms	*Haplotaxida*			
Centipedes	*Chilopoda clade*			
Snails	*Gastropoda class*			
**Fish**				
Unknown fish sp.	-	N	-	6
**Amphibian**				
Kunapalari (wheatbelt) frog ^E^	*Neobatrachus kunapalari*	N	16	1
Western banjo frog ^E^	*Limnodynastes dorsalis*	N	19	1
**Reptile**				
South-western clawless gecko ^E^	*Crenadactylus ocellatus*	N	1	2
Fine-faced gecko	*Diplodactylus pulcher*	N	4	2
Tree dtella	*Gehyra variegata*	N	4	6
Bynoe’s gecko	*Heteronotia binoei*	N	2	1
Leopard skink	*Ctenotus patherinus*	N	8	3
Unknown skink sp.	*Ctenotus* sp.	N	7	3
King’s skink ^E^	*Egernia kingii*	N	288	1
Southwestern earless skink	*Hemiergis initialis*	N	1	1
Unknown skink sp.	*Morethia* sp.	N	1	1
Bobtail skink ^E^	*Tiliqua rugosa rugosa*	N	512	1
Unknown skink sp.	Family: Scincidae	N	1	2
Western heath dragon	*Ctenophorus adelaidensis*	N	2	1
Ornate crevice dragon	*Ctenophorus ornatus*	N	2	1
Western netted dragon	*Ctenophorus reticulatus*	N	2	1
Western bearded dragon ^E^	*Pogona minor minor*	N	31	5
Thorny devil	*Moloch horridus*	N	38	1
Gould’s monitor	*Varanus gouldii*	Y	350	1
Black-headed monitor	*Varanus tristis*	Y	200	1
Unknown monitor sp.	*Varanus* sp.	Y	-	3
Southern blindsnake	*Anilios australis*	N	4.5 ?	1
Dark-spined blindsnake	*Anilios bicolor*	N	4.5	1
Pale-headed blindsnake ^E^	*Anilios hamatus*	N	20	1
Unknown blindsnake sp.	*Anilios* sp.	N	20	2
Southern shovel-nosed snake	*Brachyurophis semifasciata*	N	100	1
Bardick snake	*Echiopsis curta*	Y	400	1
Western crowned snake ^E^	*Elapognathus coronatus*	Y	40	1
Gould’s hooded snake ^E^	*Parasuta gouldii*	Y	200	1
Western brown snake	*Pseudonaja mengdeni*	Y	700	1
Unknown snake sp.	Family: Elapidae	Y	-	2
Unknown reptile sp.	-	-	-	1
**Bird**				
Domestic chicken *^,^^ƚ^	*Gallus domesticus*	N	1000	8
Laughing dove *	*Streptopelia senegalensis*	N	82	1
Spotted dove	*Streptopelia chinensis*	N	132	1
Unknown dove sp.	Family: Columbidae	N	155	2
Brush bronzewing	*Phaps elegans*	N	200	2
Australian wood duck	*Chenonetta jubata*	N	450	3
Stubble quail	*Coturnix pectoralis*	N	115	3
Unknown button-quail sp.	*Turnix* sp.	N	110	1
Galah	*Eolophus roseicapillus*	Y	300	7
Rainbow lorikeet *	*Trichoglossus haematodus*	Y	116	1
Purple-crowned lorikeet	*Glossopsitta porphyrocephala*	Y	44	1
Western rosella ^E^	*Platycercus icterotis*	Y	61	3
Red-capped parrot ^E^	*Purpureicephalus spurius*	Y	116	1
Australian ringneck parrot	*Barnardius zonarius*	Y	137	33
Elegant parrot	*Neophema elegans*	Y	43	1
Unknown parrot sp.	Order: Psittaciformes	Y	149	1
Unknown non-passerine sp.	-	N	-	2
Rufous tree creeper	*Climacteris rufa*	N	31	1
Unknown fairy wren sp.	*Malurus* sp.	N	10	1
New Holland honeyeater	*Phylidonyris novaehollandiae*	N	20	2
White-cheeked honeyeater	*Phylidonyris niger*	N	20	1
Unknown honeyeater sp.	*Phylodonyris* sp.	N	20	1
Yellow-throated miner	*Manorina flavigula*	N	55	1
Rufous whistler	*Pachycephala rufiventris*	N	25	1
Australian raven	*Corvus coronoides*	N	650	1
Willie wagtail	*Rhipidura leucophrys*	N	20	1
Grey fantail	*Rhipidura albiscapa*	N	9	1
Magpie-lark	*Grallina cyanoleuca*	N	83	1
Unknown small passerine sp.	-	N	-	3
Unknown bird sp.	-	-	-	22
**Mammal**				
European rabbit *	*Oryctolagus cuniculus*	Y	1800	34
House mouse *	*Mus musculus*	N	19	167
Black rat *	*Rattus rattus*	Y	218	21
Western bush rat ^E^	*Rattus fuscipes fuscipes*	Y	76	1
Fat-tailed dunnart	*Sminthopsis crassicaudata*	N	15	1
Quenda (bandicoot) ^E,CS^	*Isoodon fusciventer*	N	755	1
Western pygmy possum ^E^	*Cercartetus concinnus*	N	45	1
Common brushtail possum ^E^	*Trichosurus vulpecula hypoleucus*	Y	2850	10
Black-footed rock wallaby ^E,CS^	*Petrogale lateralis*	Y	4050	1
Unknown mammal sp.	-	-	-	1
Total number of cats eating vertebrates				256
Grand total cats eating invertebrates and vertebrates				294

* Introduced species in WA. ^E^ Endemic WA species. ^CS^ Species is of conservation significance. ? Body mass estimated based on that for congeneric species. ^ƚ^ Species that were not included in the analysis of prey mass predictors as they could represent scavenging of carrion.

**Table 5 animals-10-00707-t005:** Evidence of feral cats (*Felis catus*) killing medium-sized Australian mammals (1 kg or larger); all these species are of conservation significance. For reference, the body mass of cats in our studies averaged 3.08 ± 1.24 kg, *n* = 568. Unless given by specific publications (e.g., Page et al. [83]), body mass measures of prey species are from Van Dyck and Strahan [28].

Prey Species	Average Adult Body Mass (by Sex)	Evidence	Adult/Juvenile	Reference
Tammar wallaby (*Macropus eugenii*)	6.0 kg	Extirpation from islands	J and A	Dickman [7]
Tasmanian pademelon (*Thylogale billardierii*)	M 7.0 kg (3.8–12.0 kg) F 3.9 kg (2.4–10.0 kg)	Camera trap	A (F ≈ 4 kg)	Fancourt [81]
Allied rock-wallaby (*Petrogale assimilis*)	M 4.7 kg F 4.3 kg	Direct observation, carcasses *	J and A (≈4.0 kg)	Spencer [84]
Black-footed rock wallaby (*Petrogale lateralis*)	M 4.5 kg (4.1–5.0 kg) F 3.5 kg (3.1–3.8 kg)	Stomach contents	J	This study
		Stomach contents and direct observation of cat feeding at a freshly-killed adult	J and A	Read et al. [44]
		Stomach contents	?	Paltridge et al. [57]
Bridled nailtail wallaby (*Onychogalea fraenata*)	M 6.0 kg (5.0–8.0 kg) F 4.5 kg (4.0–6.0 kg)	Stomach contents	J (≈1.5 kg)	Horsup and Evans [85]
		Predation ƚ	J (≈3.0 kg)	Fisher, Blomberg and Hoyle [86]
Spectacled hare-wallaby (*Lagorchestes conspicillatus*)		Extirpation from islands	J and A	Dickman [7]
Rufous hare-wallaby (*Lagorchestes hirsutus*)	M 1.6 kg (1.2–1.8 kg) F 1.7 kg (0.8–2.0 kg)	Stomach contents Predation ƚ	? J and A	Gibson et al. [82]
		Predation ƚ	A	Hardman et al. [87]
		Stomach contents	?	Paltridge et al. [57]
Banded hare-wallaby (*Lagostrophus fasciatus*)	1.6 kg (1.0–2.3 kg)	Predation ƚ	A	Hardman et al. [87]
Brush-tailed bettong/woylie (*Bettongia penicillata*)	M 1.27 kg (0.98–1.85 kg) F 1.40 kg (0.75–1.50 kg)	Predation ƚ Loss off islands	A	Marlow et al. [88] Dickman [7]
Burrowing bettong/boodie (*Bettongia lesueur*)		Loss off islands		Dickman [7]

* Carcasses showing characteristic evidence of having been eaten by a cat. ƚ Predation of radio-collared animals. ? No way to identify whether the animals were taken were juvenile or adult from analysis of hair in stomach contents.

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
