# Peer review of "Body Size and Bite Force of Stray and Feral Cats—Are Bigger or Older Cats Taking the Largest or More Difficult-to-Handle Prey?"

_animals, 2020, doi:10.3390/ani10040707_

Round 1
Reviewer 1 Report
The study is well done but the conclusions are not particularly exciting. Nevertheless, we need more data on cat predation patterns, especially in Australasia and the paper merits publication for this reason.
Comments on Fleming et al, Predation by Feral Cats. Animals 723824
Line 110 – “pes” is an unusual term – definition is suggested.
Line 164 – I would just note that I do not have the expertise to review the statistical methods or the assumptions underlying bite force calculation.
Line 170 – I do not see any discussion on how BCI is evaluated for a frozen carcass.
Line 194 – There is no table indicating the numbers of animals analysed. The paper reports that 568 carcasses were recovered but that the stomach contents were analyzed for only 352 and that skull examination could be done for only 268. I would suggest including a table indicating the n for gender, age, location for the 352 and for the 268 carcasses. It might be useful to include a note about mode of death (I presume 84 of the cats analysed were shot since only 268 skulls out of 352 carcasses could be analysed.
Line 206 – Data on sample composition in this section.
Line 226 – Did stray cats have lower body mass than the feral cats? That might be a valuable point to enlarge on. Stray cats have access to less food resources than feral cats?
Line 474 – Seems overly alarmist to suggest from a single case that cat predation is a “significant” conservation issue for medium-sized marsupials! In general, the biggest conservation threat to wildlife in most of the world is human expansion and transformation of wild places. Cat predation is an extra threat that must be addressed for vulnerable marsupial populations, but they are now vulnerable or endangered primarily due to human expansion into natural ecosystems.
Line 525 – How does one define refuse as a food item if not as “carrion”? Are cats put off by chemicals in decomposing flesh?
Author Response
Reviewer 1
The study is well done but the conclusions are not particularly exciting. Nevertheless, we need more data on cat predation patterns, especially in Australasia and the paper merits publication for this reason.
We were obviously disappointed to not have data with strong predictive power, but the key message is that even the young/small cats can be deadly.
Line 110 – “pes” is an unusual term – definition is suggested.
Definition added.
Line 164 – I would just note that I do not have the expertise to review the statistical methods or the assumptions underlying bite force calculation.
Thank you for your honesty.
Line 170 – I do not see any discussion on how BCI is evaluated for a frozen carcass.
Line 105 – the carcasses were thawed before body mass and body size measurements were recorded.
Line 194 – There is no table indicating the numbers of animals analysed. The paper reports that 568 carcasses were recovered but that the stomach contents were analyzed for only 352 and that skull examination could be done for only 268. I would suggest including a table indicating the n for gender, age, location for the 352 and for the 268 carcasses.
Table 1 includes the numbers that you refer to, broke down by sex and by location.
It might be useful to include a note about mode of death (I presume 84 of the cats analysed were shot since only 268 skulls out of 352 carcasses could be analysed.
We have added such a note. This is highlighted in the text.
Line 206 – Data on sample composition in this section.
Apologies if we do not understand this comment, but the percentages for the 108 females and the 160 male cats are given in this sentence.
Line 226 – Did stray cats have lower body mass than the feral cats? That might be a valuable point to enlarge on. Stray cats have access to less food resources than feral cats?
Table 1 includes the results of a Mann-Whitney U test comparing the body mass for rural and urban cats – there was no significant difference in their body mass. The first paragraph of the results states such: “Although rural cats sampled were marginally smaller than urban cats, only the head lengths were significantly different; there was no significant effect of sample location on the mass or head-body length between source locations”.
Line 474 – Seems overly alarmist to suggest from a single case that cat predation is a “significant” conservation issue for medium-sized marsupials! In general, the biggest conservation threat to wildlife in most of the world is human expansion and transformation of wild places. Cat predation is an extra threat that must be addressed for vulnerable marsupial populations, but they are now vulnerable or endangered primarily due to human expansion into natural ecosystems.
Review of the literature for other instances of prey being taken by cats is presented in Table 5. We believe that, taken together, there is sufficient indication that cats are taking medium-sized prey.
Line 525 – How does one define refuse as a food item if not as “carrion”? Are cats put off by chemicals in decomposing flesh?
We have included a description of the material that was considered refuse in our methods section: “Refuse items were classified as sources of food that would have been taken from anthropogenic sources and included meat scraps, vegetable and fruit scraps, paper, plastic, aluminium foil, packing styrofoam, glass shards, synthetic fibres etc.”
Reviewer 2 Report
This is a fantastic and almost flawless manuscript. I could not really find any errors or conceptual issues. Statistics are done simply and right, results are presented well. My only comments are:
Paragraph 356, avoid using 'interestingly', as you dont decide that, the reader does. Also, you explain why there is no strong relationship between bite force vs condition well, making it no longer interesting. Again, line 382, dont use 'interestingly'. I dont think it is interesting cats dont eat carrion, it is well established on mainland Australia.
Line 459: "Although prey less than 200 g are usually preferred by cats" Those references do not really state prey preference, more prey occurrence. If a cat had the choice between a 300g bandicoot or 50g hopping mouse, i am sure they would prefer the bandicoot. Those references (Dickman 1996 and Doherty et al 2015) do not properly take into account prey availability at all sites.
Para 559. This paragraph needs a better opener to frame this in your study. You spend around 8 lines talking about gene drives, leaving the reader wondering why for a long time. I suggest add something like "our research highlights potential issues with gene drives".
Author Response
Reviewer 2
This is a fantastic and almost flawless manuscript. I could not really find any errors or conceptual issues. Statistics are done simply and right, results are presented well.
Thank you!
My only comments are:
Paragraph 356, avoid using 'interestingly', as you dont decide that, the reader does. Also, you explain why there is no strong relationship between bite force vs condition well, making it no longer interesting. Again, line 382, dont use 'interestingly'. I dont think it is interesting cats dont eat carrion, it is well established on mainland Australia.
Very fair. We have removed these words.
Line 459: "Although prey less than 200 g are usually preferred by cats" Those references do not really state prey preference, more prey occurrence. If a cat had the choice between a 300g bandicoot or 50g hopping mouse, i am sure they would prefer the bandicoot. Those references (Dickman 1996 and Doherty et al 2015) do not properly take into account prey availability at all sites.
Point taken about prey preference vs occurrence. We have modified the text in this paragraph.
Para 559. This paragraph needs a better opener to frame this in your study. You spend around 8 lines talking about gene drives, leaving the reader wondering why for a long time. I suggest add something like "our research highlights potential issues with gene drives".
We have added a better lead-in to this material – hopefully the point is now more clear.